# The visual pigment xenopsin is widespread in protostome eyes and impacts the view on eye evolution

**Clemens Christoph Döring[†], Suman Kumar[†], Sharat Chandra Tumu, Ioannis Kourtesis, Harald Hausen\***

Sars International Centre for Marine Molecular Biology, University of Bergen, Bergen, Norway

**Abstract** Photoreceptor cells in the eyes of Bilateria are often classified into microvillar cells with rhabdomeric opsin and ciliary cells with ciliary opsin, each type having specialized molecular components and physiology. First data on the recently discovered xenopsin point towards a more complex situation in protostomes. In this study, we provide clear evidence that xenopsin enters cilia in the eye of the larval bryozoan *Tricellaria inopinata* and triggers phototaxis. As reported from a mollusc, we find xenopsin coexpressed with rhabdomeric-opsin in eye photoreceptor cells bearing both microvilli and cilia in larva of the annelid *Malacoceros fuliginosus*. This is the first organism known to have both xenopsin and ciliary opsin, showing that these opsins are not necessarily mutually exclusive. Compiling existing data, we propose that xenopsin may play an important role in many protostome eyes and provides new insights into the function, evolution, and possible plasticity of animal eye photoreceptor cells.

## Introduction

**\*For correspondence:**
harald.hausen@uib.no

[†]These authors contributed equally to this work

**Competing interests:** The authors declare that no competing interests exist.

The photoreceptor cells (PRCs) in animal eyes are often classified according to their structure, that is depending on whether the sensory surface is enlarged by microvilli or by cilia (*Eakin, 1979*; *Eakin, 1963*; *Eakin, 1968*). The first type of PRCs in many protostomes was shown to depolarize in response to light and to employ rhabdomeric opsin (r-opsin) as a visual pigment, which signals via the $G\alpha_q$ mediated $IP_3$ cascade opening TRP ion channels in the PRC membrane (*Fain et al., 2010*; *Shichida and Matsuyama, 2009*). In contrast, ciliary PRCs of vertebrate eyes are known to signal via the $G\alpha_{i/t}$ mediated cGMP cascade closing CNG channels and leading to a hyperpolarization. Since both are found in protostome and deuterostome animals and due to their distinct molecular signatures, it is assumed that these two kinds of PRCs were already present in the last common ancestor of bilaterian animals (*Arendt, 2008*; *Arendt et al., 2004*; *Arendt et al., 2002*; *Gomez et al., 2009*; *Gehring, 2014*; *Nasi and Gomez, 2009*; *Panda et al., 2002*). This classification of PRCs became a sound basis for comparative eye research from sensory biology to molecular physiological, developmental, and evolutionary biology. We present data suggesting that in protostomes, an additional second kind of ciliary PRCs is widespread and that this may be evolutionarily closer to microvillar PRCs than to vertebrate ciliary eye PRCs.

Recently, a new type of visual opsins, xenopsin, has been characterized. It shares important functional sequence motifs with ciliary opsins (c-opsins) and has been shown to signal most likely also via $G\alpha_i$ in a flatworm (*Rawlinson et al., 2019*). Nonetheless, xenopsins and c-opsins do not group in phylogenetic analyses (*Ramirez et al., 2016*; *Rawlinson et al., 2019*; *Vöcking et al., 2017*) indicating a distinct evolutionary origin. Surprisingly, xenopsins and c-opsins are mutually exclusively distributed across the animal kingdom, which is difficult to explain from a genomic perspective and seemingly doubts the phylogenetic analyses. In this study, we report the first organism having both

xenopsin and c-opsin. In congruence with thorough phylogenetic and gene structure analyses, this provides further support for a distinct evolutionary origin of these visual pigments.

Despite increasing knowledge on the presence of xenopsin in many animal groups, only very few data on cellular expression and function of xenopsin exist. So far it turned out to be this new opsin type and not c-opsin that is present in ciliary eye PRCs of larval brachiopods (*Passamaneck et al., 2011*; *Vöcking et al., 2017*) and in larval ciliary eye PRCs and adult extraocular ciliary PRCs in a flatworm (*Rawlinson et al., 2019*). Furthermore, xenopsin has been found coexpressed with r-opsin in eye PRCs exhibiting both microvilli and cilia in the larva of a mollusc (*Vöcking et al., 2017*), thereby raising the question, whether protostome eye PRCs had the potential to change between microvillar and ciliary organization during evolution.

To obtain a broader overview of the role of xenopsin in animal eyes, we investigated larva of the annelid *Malacoceros fuliginosus* (*Claparède, 1868*), and the bryozoan *Tricellaria inopinata* d'Hondt & Occhipinti Ambrogi, 1985 in which RNA-seq data pointed to the presence of xenopsin. We find it expressed in ciliary eye PRCs of the bryozoan larva, and we present unambiguous evidence that xenopsin enters the cilia and likely triggers the phototactic response of the larva. Further, we find xenopsin coexpressed with r-opsin in eye PRCs of the annelid larva similar to the earlier finding in a larval chiton (*Vöcking et al., 2017*). We propose that (1) Xenopsin is an important visual pigment in protostomes, (2) ciliary eye PRCs may not be of the same evolutionary origin in protostomes and deuterostomes, and (3) ciliary and microvillar eye PRCs may be evolutionarily linked in protostomes. The findings impact the current understanding of how animal eyes evolved and diversified and provides insights on the plasticity that cell types can exhibit in the course of evolution.

## Results

### Molecular phylogeny of animal xenopsins and c-opsins

We screened *Tricellaria inopinata* assembly one for opsins by blasting with a broad set of metazoan opsin sequences as query and successive reciprocal blast against Genbank. The sequences were further checked for the presence of the PFAM 7tm_1 domain and the residue Lys296, which is predictive for chromophore binding in opsins and for the NPXXY motif at positions 302–306 (*Figure 1*) contributing to signal transduction in G protein-coupled receptors. We blasted the hits against assembly two and elongated the sequences if longer hits were retrieved. We screened the transcriptomic resources of *Malacoceros fuliginosus* in the same manner, but only for the presence of xenopsins and c-opsins. We retrieved five hits from the *T. inopinata* assembly, which all gave xenopsins as first hits by reciprocal blast. Since we had evidence for contamination of the *T. inopinata* assembly (see Materials and methods), we cloned all sequences and tested them by ISH for expression in *T. inopinata* larva. Only one sequence gave positive signals and was further used in this study, while the others were no longer considered as they might be from other bryozoan species. Three sequences were retrieved from the *M. fuliginosus* assembly. After reciprocal blast against Genbank, one sequence gave c-opsins as first hits and the other two xenopsins. For further analyses, we kept the potential c-opsin and that potential xenopsin, for which we obtained positive results after in situ hybridization in larvae.

We added the sequences and few recently described xenopsin sequences from the molluscs *Sepia officinalis* and *Ambigolimax valentianus*, the bryozoan *Bugula neritina*, the flatworm *Maritigrella crozieri*, and the chaetognath *Pterosagitta draco* to the opsin sequence set (https://doi.org/10.7554/eLife.23435.009) analyzed by *Vöcking et al., 2017* and ran maximum likelihood and Bayesian phylogenetic analyses to study opsin molecular evolution with a focus on the relationships of xenopsins and c-opsins. All major opsin groups described by *Ramirez et al., 2016*, *Vöcking et al., 2017*, and *Rawlinson et al., 2019* such as tetraopsins, r-opsins, cnidops, ctenopsins, c-opsins, and xenopsins were recovered with high support values (*Figure 2*, *Figure 2—figure supplements 1* and *2*). One sequence from *M. fuliginosus* falls into c-opsins, while another one falls into xenopsins. The opsin of *T. inopinata* likewise falls into xenopsin and groups with the sequence of the bryozoan *Bugula neritina*. The topology within the xenopsin clade suggests an early divergence of xenopsin in two clades xenopsin A and xenopsin B containing opsin from several animal groups similar as described by *Vöcking et al., 2017* and *Rawlinson et al., 2019*. Yet, the support values for the two subclades are not as high as for xenopsisn as a whole and other large opsin groups. We tested

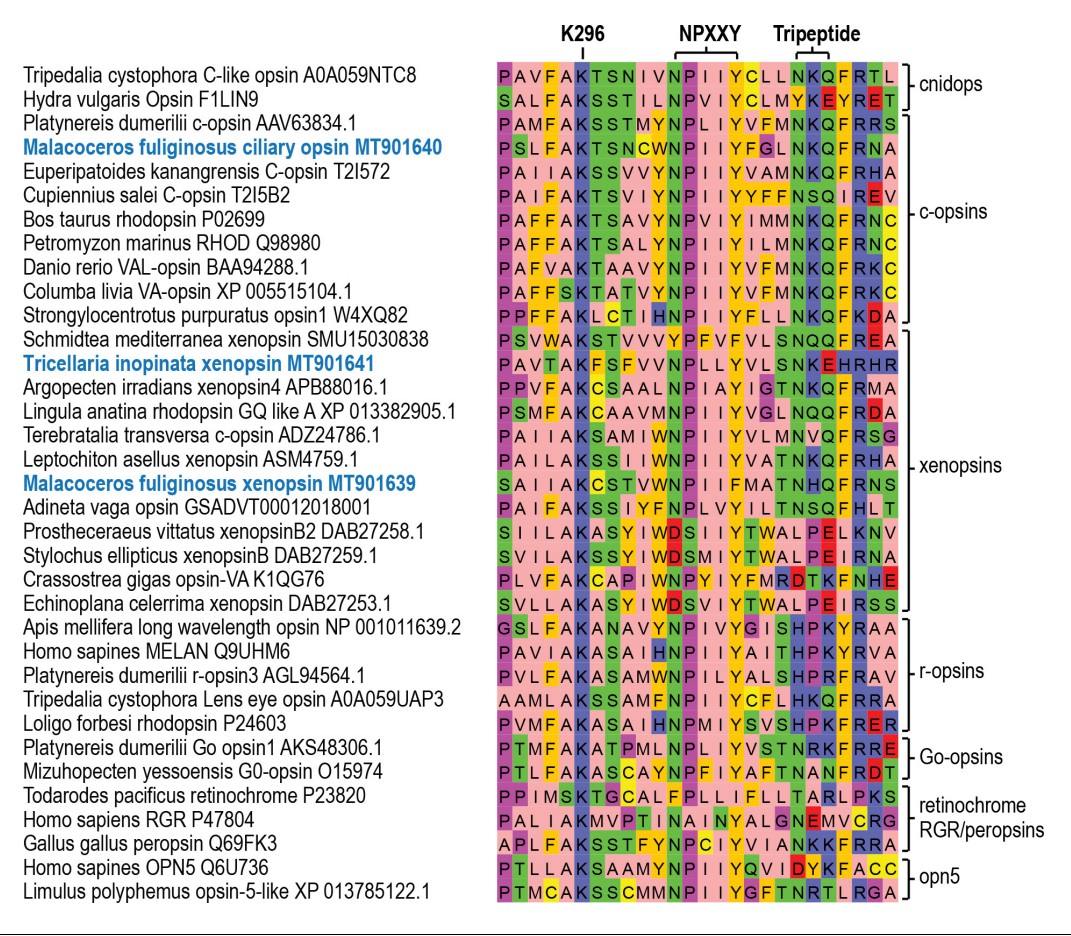

**Figure 1.** Conservation of functionally important motifs and residues in different opsin types. Alignment of parts of the transmembrane domain VII and the cytosolic helix VIII of selected opsin sequences showing the conserved lysine 296 (K296) chromophore binding site and other conserved motifs important for opsin-G protein interaction like NPXXY and the tripeptide (NKQ in c-opsins and several xenopsins; HPK in r-opsins). The sequences investigated in this study are highlighted in blue.

The online version of this article includes the following figure supplement(s) for figure 1:

**Figure supplement 1.** Conservation of functionally important motifs and residues in xenopsins.

robustness of the split into xenopsin A and B against changes in the outgroup by calculating trees of xenopsins only (unrooted) and trees with few ciliary opsins, few cnidops and few c-opsins and cnidops as outgroup. The split is retained in all cases with the exception of an outgroup composed of cnidops and c-opsins, where xenopsin B is a paraphyletic assemblage (*Figure 2—figure supplements 5–8*). The position of one brachiopod sequence (Lingula anatina melanopsin like XP 013397676.1) is not stable, in some cases it falls into xenopsin A, in others it groups with xenopsin B sequences or has a basal position. Accordingly, our data suggest an early diversification of xenopsins, but with moderate support only. Since *M. fuliginosus* xenopsin groups in all trees with xenopsin B representatives, we regard it as likely being the first known annelid xenopsin B. Several flatworm xenopsin B sequences stand out by strong modifications in the NPXXY and tripeptide motif (*Figure 1—figure supplement 1*) questioning the capability of the opsins to induce G-protein based light transduction. In difference, these motifs are conserved in the xenopsin of *M. fuliginosus*.

## Gene structure analysis corroborates molecular phylogeny

Several sequences (for example from *Idiosepius paradoxus* and *Terebratalia transversa*), which in *Ramirez et al., 2016*, *Vöcking et al., 2017*, *Rawlinson et al., 2019* and this study group within xenopsins were earlier classified as c-opsins (*Passamaneck et al., 2011*; *Yoshida et al., 2015*). This view

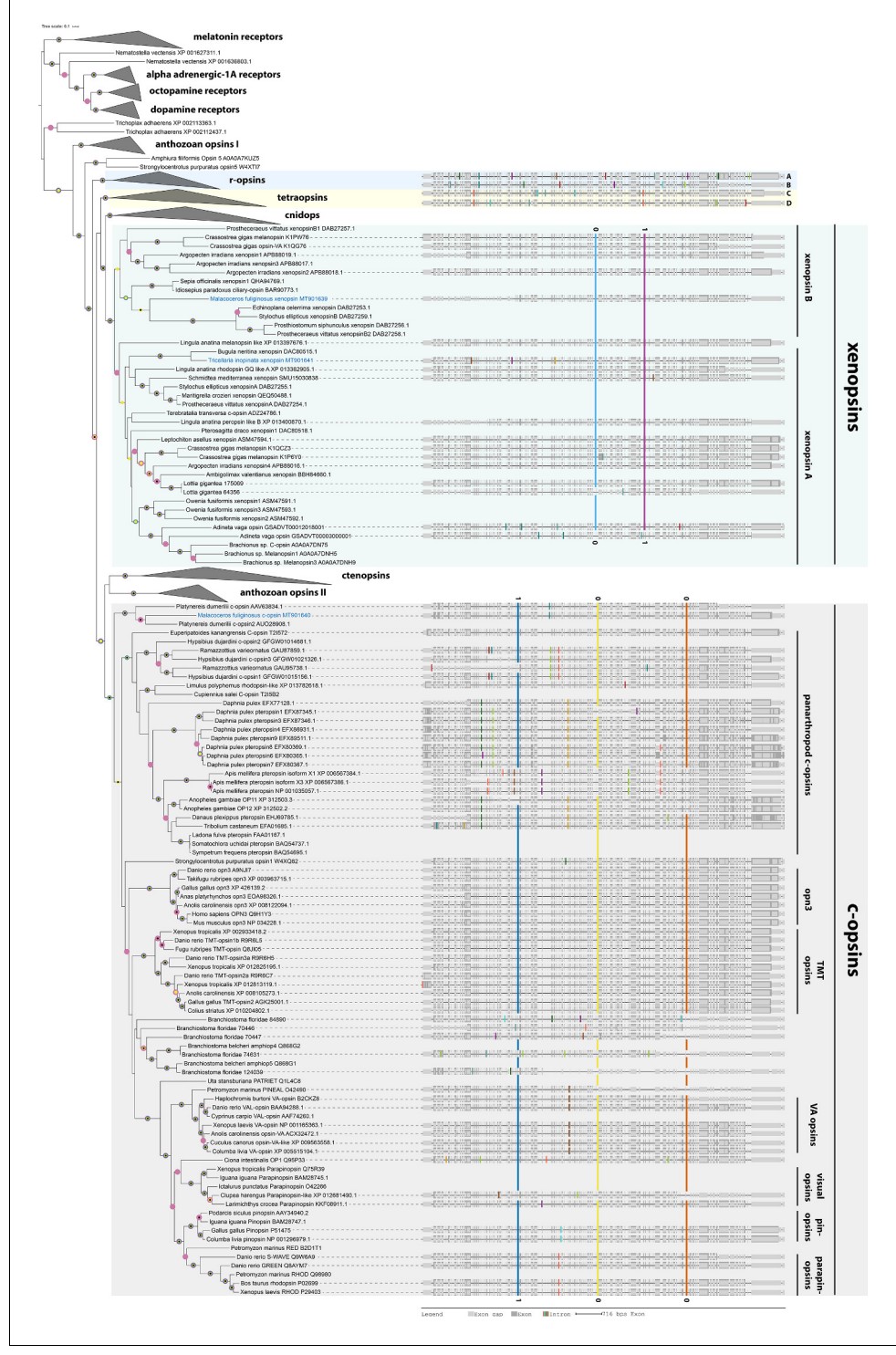

**Figure 2.** C-opsins and xenopsins display type-specific conserved gene structures. Maximum Likelihood tree of opsin protein sequences (IQ-TREE, LG+F+R8). Labeled nodes have support values of SH-like approximate likelihood ratio test (blue dot) and ultrafast bootstrap ≥0.9 (purple dot), approximate Bayes test ≥0.98(yellow dot), and a posterior probability ≥0.95 (black dot) in a parallel Bayesian analysis (Phylobayes, DS-GTR + G, consensus of two out of three chains, 90,000 cycles). Intron positions (colored bars) are mapped on the un-curated protein sequence alignment, and introns conserved in position and phase are highlighted by bars spanning several sequences and labels for the intron position. The sequences investigated in this study are highlighted in blue. The xenopsins of *M. fuliginosus* and *T. inopinata* display xenopsin type gene structures. The c-opsin of *M. fuliginosus*

*Figure 2 continued on next page*

*Figure 2 continued*

groups with *Platynereis dumerilii* c-opsin going along with a corresponding gene structure. For r-opsins and tetraopsins gene structures are shown for A: Homo sapiens MELAN Q9UHM6, B: *Apis mellifera* UV opsin AAC47455.1, C: *Limulus polyphemus opsin-5-like* XP 013785122.1, and D: Homo sapiens OPN5 Q6U736. See *Figure 2—figure supplement 1* for un-collapsed ML tree, *Figure 2—figure supplement 2* for un-collapsed Phylobayes tree, *Figure 2—figure supplement 3* for the whole set of gene structures, *Figure 2—figure supplement 4* for intron phases, *Figure 2—figure supplement 5* for an unrooted tree of only xenopsins, *Figure 2—figure supplement 6* for a tree of only xenopsins plus a few c-opsins as outgroup, *Figure 2—figure supplement 7* for a tree of xenopsins only plus a few cnidops as outgroup, *Figure 2—figure supplement 8* for a tree of xenopsins only plus a few c-opsins and cnidops as outgroup and *Figure 2—source data 1* for gene accession numbers.

The online version of this article includes the following source data and figure supplement(s) for figure 2:

**Source data 1.** Accession numbers of the genes used for gene tree inference.
**Figure supplement 1.** Un-collapsed tree of phylogeny shown in *Figure 2*.
**Figure supplement 2.** Bayesian analysis (Phylobayes) of sequence alignment used in *Figure 2*.
**Figure supplement 3.** Gene structures of all sequences, which were used for gene tree calculation and for which genomic information was available or generated in this study, mapped on the un-curated protein sequence alignment.
**Figure supplement 4.** Intron phase and position of all sequences, which were used for gene tree calculation and for which genomic information was available or generated in this study, mapped on the un-curated protein sequence.
**Figure supplement 5.** Unrooted xenopsin tree.
**Figure supplement 6.** Xenopsin tree rooted with few c-opsins.
**Figure supplement 7.** Xenopsin tree rooted with few cnidops.
**Figure supplement 8.** Xenopsin tree rooted with few c-opsins and cnidops.

was either based on automated gene annotation, similarity searches, or phylogenetic analyses with only low taxon sampling. Nonetheless, it is in congruence with the presence of the NKQ tripeptide pattern (*Figure 1*) in the fourth cytoplasmic loop, which is in c-opsins crucial for specific binding to $G\alpha_{i/t}$ (*Marin et al., 2000*). To test if the grouping of the new opsin sequences found in *T. inopinata* and *M. fuliginosus* may result from tree inference artifacts — we cloned the respective genes from genomic DNA, analyzed gene structure and mapped it together with gene structure data generated by *Vöcking et al., 2017* onto the protein alignment. Both the xenopsin and the c-opsin groups have specific gene structures. Three introns are highly conserved in position and intron phase throughout c-opsins. In comparison, two distinct introns in xenopsins are conserved likewise in position and intron phase (*Figure 2*, *Figure 2—figure supplements 3* and *4*). The gene structures of the new *T. inopinata* and *M. fuliginosus* xenopsins and the *M. fuliginosus* c-opsin match well with those of other xenopsins and c-opsins, respectively, and strongly corroborates the classification based on the molecular phylogenetic analysis. The xenopsin clade contains only sequences from protostomes. Notably, the closest related clade is neither a deuterostome specific nor a protostome specific opsin group, but cnidops. Yet, validation of this sister group relationship by gene structure data is not possible, since cnidops are lacking introns.

## Xenopsin is expressed in cilia of the eye photoreceptor cells in larval *T. inopinata*

Larvae of *T. inopinata* possess one median eye apical of the anterior vibratile plume and one pair of lateral eyes halfway down from the apical to the abapical pole (*Figure 3A*). All eyes can be easily identified in live animals due to their red pigmentation. EM sections show that all three eyes form epidermal invaginations (*Figure 4A*, *Figure 4—figure supplements 1A* and *2*), and whole-mount in situ hybridization revealed that Tin-xenopsin is strongly expressed in the region of all three eyes (*Figure 3B,C*). Besides, we found weak expression of Tin-xenopsin in few other cells, which are not associated with shielding pigments. One pair of cells lies on the rim of the anterior ciliary groove. Another pair lies lateral to the axial nerve running down from the apical organ (*Figure 3B,C*). These cells have small projections that also show weak expression of Tin-xenopsin (*Figure 3—figure supplement 1A,B*) and extend towards the body surface. A third pair lies lateral to the opening of the internal sac at the abapical pole (*Figure 3B,C*, *Figure 3—figure supplement 1C*). The in situ

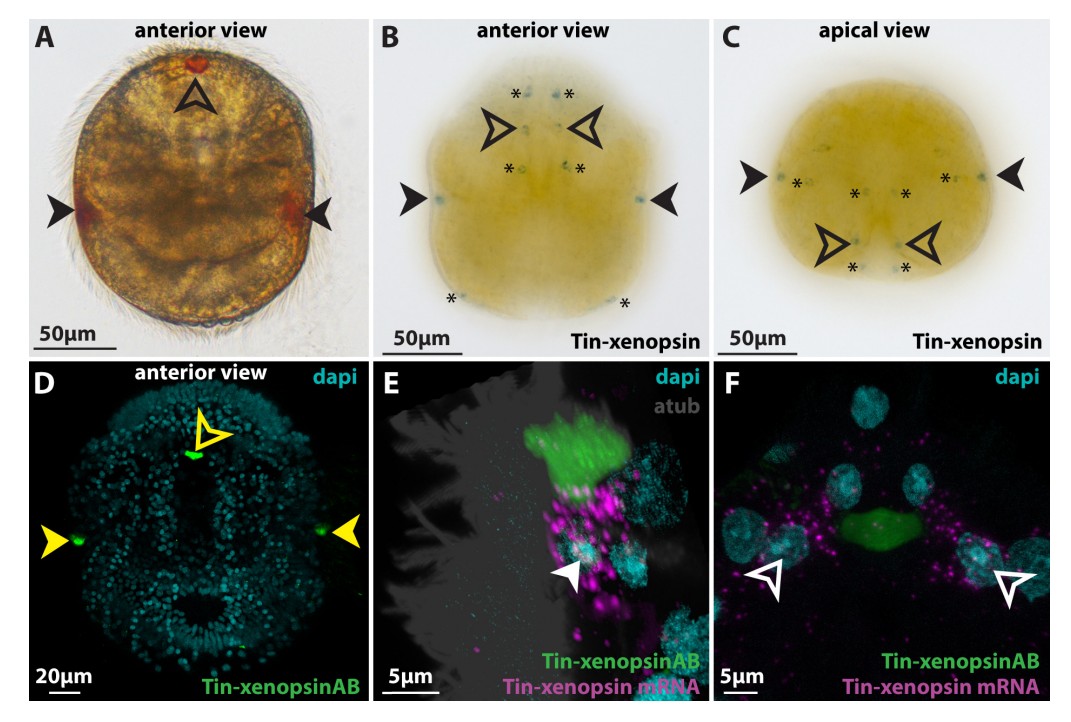

**Figure 3.** Xenopsin expression in *Tricellaria inopinata*. (A) Anterior view of a larva showing the pigment spots of the paired lateral eyes (filled arrowheads) and the single median eye (outlined arrowheads). (B,C) WMISH of Tin-xenopsin. Maximum projections of z-stacks spanning the whole larva. Single spots are labeled in the positions of the lateral eyes and two spots in the position of the single median eye. Several cells not associated with shielding pigment (asterisks) are also labeled. (D) Anti Tin-xenopsin antibody labels only the eyespot regions (filled yellow arrowheads: lateral eyes, outlined yellow arrowhead: median eye). (E,F) Combination of ISH and IHC. (E) Lateral eye. Tin-xenopsin antibody localizes adjacent to the mRNA around the nucleus of the eye photoreceptor cell (filled white arrowhead). (F) Median eye. Tin-xenopsin antibody localizes between a left and a right photoreceptor cell (outlined white arrowheads). See *Figure 3—figure supplement 1* for details on Tin-xenopsin expression in extraocular cells. The online version of this article includes the following figure supplement(s) for figure 3:

**Figure supplement 1.** Expression of Tin-xenopsin in cells not associated with shielding pigment (asterisks).

hybridization signal was much stronger in the eye regions than in the extraocular cells. Custom made antibodies against Tin-xenopsin specifically stain the eye regions (*Figure 3D*), but no significant staining appeared in the extraocular Tin-xenopsin expressing or any other cells.

To get insights into the fine structure of the eyes, we performed serial section electron microscopy. The invagination of the lateral eyes is 5 µm deep, and it is formed by two neighboring coronal epidermal cells (PCC1 and PCC2 in *Figure 4* and *Figure 4—figure supplement 2*) and the eye photoreceptor cell (PRC in *Figure 4* and *Figure 4—figure supplement 2*). The coronal cells differ from adjacent coronal cells by the lack of cilia and the presence of abundant shielding pigment vesicles in the region of the eye invagination. The vesicles show high electron density in the chemically fixed specimen, but moderate electron density in the cryo-fixed specimen (*Figure 4B*, *Figure 4—figure supplement 3*). The two coronal cells line the apical and the lateral walls of the invagination, while the eye PRC lines the bottom and the abapical wall (*Figure 4—figure supplement 2*). The sensory cell bears a very dense bundle of cilia (ciPRC in *Figure 4A,B*; *Figure 4—figure supplement 2*) extending into the eye invagination. In the right eye of the cryo-fixed specimen, we counted 170 cilia. The axonemal microtubules of the cilia are arranged in $9 \times 2$+two pattern (*Figure 4B*). The cilia of the PRC penetrate the cuticle (cu in *Figure 4A,B*) and fill most of the eye invagination. The individual cilia have a diameter of 200 nm and are around 11 µm in length, and their upper halves extend above the body surface. Accordingly, the total surface of the ciliary membranes is approximately 1170 µm$^2$. No other cell sends cilia into the invagination. The perikaryon of the eye PRC lies anterior to the base of the invagination (*Figure 4—figure supplement 2*). A single axon extends from the basal part of the sensory cell, joins the equatorial nerve ring, and runs towards the anterior

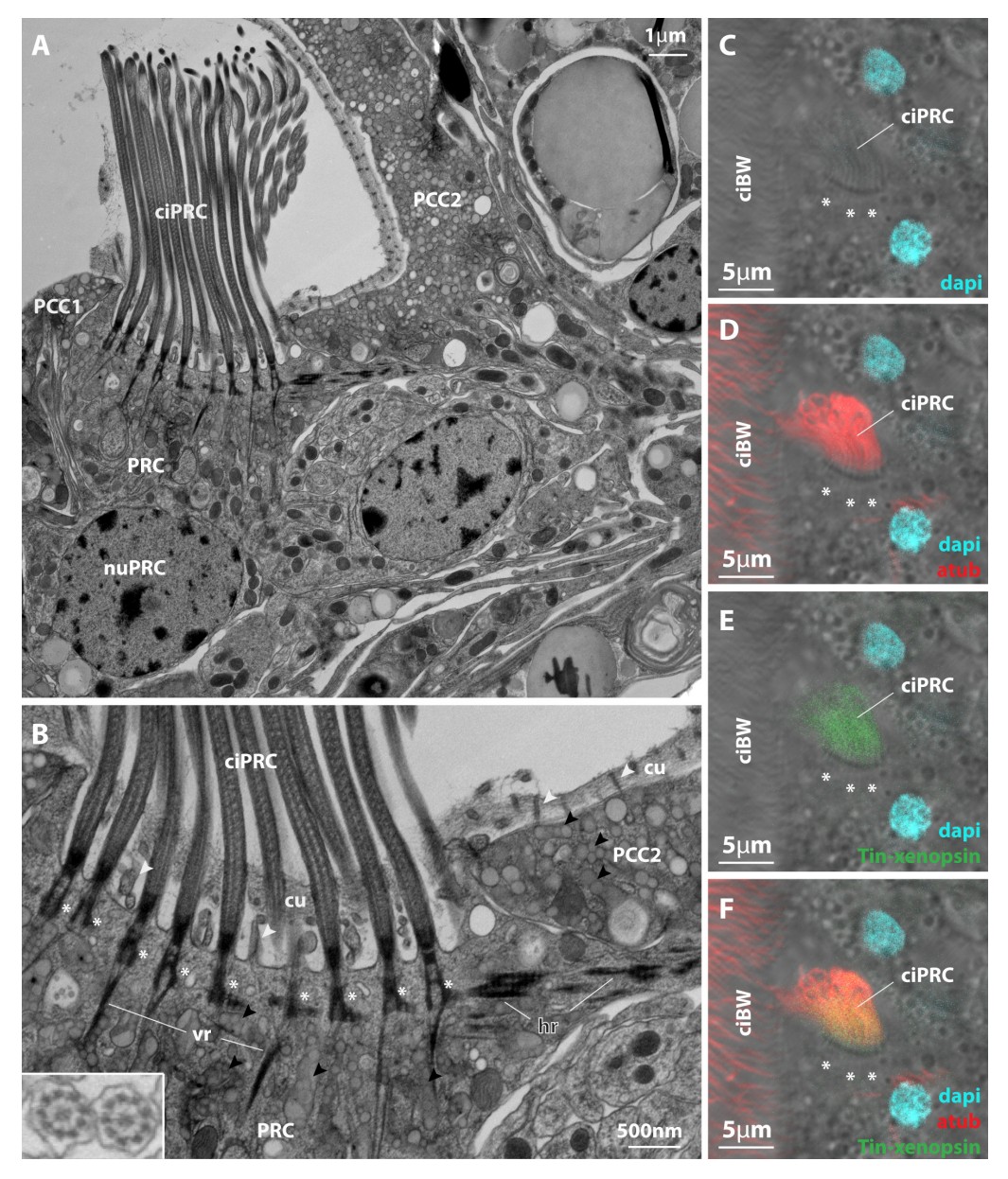

**Figure 4.** Subcellular localization of xenopsin in the lateral eye of *Tricellaria inopinata*. (A,B) Electron microscopic images (cryofixation) showing the photoreceptor cell (PRC) sending numerous cilia (ciPRC) into the eye invagination. The cilia possess basal bodies (white asterisks) and vertical (vr) and horizontal (hr) rootlets. Shielding pigment vesicles (black arrowheads) are present in the PRC and the adjacent pigmented coronal cells (PCC1, PCC2). Inset in B: cross-section of eye PRC cilia (chemical fixation) showing the 9 × 2 +2 organization of the axoneme. (C–F) IHC labeling of Tin-xenopsin and acetylated alpha-tubulin. Same orientation as in (A,B). Tin-xenopsin protein localizes within the cilia projecting into the eye invagination of the eye PRC. The basal bodies (white asterisks) are visible inside the eye PRC. ciBW: cilia of the body wall, cu: cuticle, nuPRC: nucleus of the photoreceptor cell. See *Figure 4—figure supplement 1* for Tin-xenopsin localization in the median eye, *Figure 4—figure supplement 2* for the cellular composition of the lateral eye, and *Figure 4—figure supplement 3* for differences in the appearance of shielding pigment granules between chemical and cryofixation. The online version of this article includes the following figure supplement(s) for figure 4:

**Figure supplement 1.** Subcellular localization of xenopsin in the median eye of *Tricellaria inopinata*.
**Figure supplement 2.** Organization of the lateral larval eye of *Tricellaria inopinata*.
**Figure supplement 3.** Appearance of shielding pigment granules in the lateral eye of Tricellaria inopinata.

ciliary groove. Abapical to the eye sensory cells lies an additional sensory cell (aSC in *Figure 4—figure supplement 2*). It sends a slender dendrite running upwards on the abapical side of the eye PRC and forms an anteriorly projecting pillar-like elevation emerging from the abapical wall of the eye invagination (*Figure 4—figure supplement 2*). On top of the elevation, 15 cilia with a 9 × 2 +two axoneme emerge from the tip of the dendrite and penetrate the cuticle.

The invagination of the median eye is formed by two coronal cells with shielding pigment granules and two PRCs (PRC1, PRC2 in *Figure 4—figure supplement 1A*). Subcellular characteristics of the coronal cells and PRCs are similar to those of the lateral eyes, but the cellular arrangement is different. The coronal cells line the bottom as well as the apical and abapical walls of the invagination, while the two photoreceptor cells line the left and the right wall. The perikarya of the PRCs lie distant to each other on the left and the right from the invagination and the ciliary bundles of the PRCs project from both sides into the eye invagination.

Knowing the ultrastructure of the eyes makes it possible to localize Tin-xenopsin mRNA and protein on the subcellular level by combining fluorescence in situ hybridization (FISH) with immunohistochemistry. In the lateral eye, the Tin-xenopsin FISH signal surrounds a nucleus next to the base of the eye invagination (*Figure 3E*). It matches well the position of the eye PRC nucleus in the EM dataset. The anti-Tin-xenopsin antibody signal is directly adjacent to the FISH signal and co-localizes with the cilia labeled by anti-acetylated α-tubulin in the eye invagination (*Figure 4C–F*). The median eye shows a similar pattern. While the FISH signal stains one cell on each side of the invagination, the opsin antibody stains the cilia inside the eye invagination (*Figure 3F*; *Figure 4—figure supplement 1B–E*). Accordingly, Tin-xenopsin mRNA is located throughout the soma of the eye photoreceptor cells of *T. inopinata*, whereas the opsin protein resides in the ciliary bundles emerging from these cells.

## Tin-xenopsin likely is most sensitive in blue light and triggers the phototactic response of the larva

Since we could not detect the expression of any other opsin than Tin-xenopsin in the eyes of *T. inopinata*, we were interested in behavioral responses, which depend on directional detection of light by the eyes for the first functional characterization of this new opsin. We assayed the phototactic displacement of freshly hatched larvae under different wavelengths. The animals showed the biggest displacement towards blue light (454 nm) but still showed displacement towards green (513 nm), cyan (506 nm), and purple/UV (407 nm) (*Figure 5*, *Figure 5—figure supplement 1*). We could not detect a reaction with wavelength beyond the green spectrum (593 nm, 612 nm, 630 nm).

## Xenopsin is coexpressed with r-opsin in cerebral eye PRCs in larval *M. fuliginosus*

M.*M. fuliginosus* larvae develop three pairs of pigmented eyespots in the head - one in a midventral position, one mediodorsal, and the third one in a laterodorsal position (*Figure 6A,B*). The ventral eyespots develop first at around 14 hpf, and the two pairs of dorsal eyespots develop at around 42 hpf. Preliminary investigation of ultrastructural data at 72 hpf stage revealed a mainly rhabdomeric organization of the ventral and mediodorsal eyespots, whereas the third laterodorsal eyespot revealed a ciliary structure. The ventral eyespot has three photoreceptor cells (PRCs), sending dense microvilli into the concavity made by two pigment cup cells (PCs) (*Figure 6G–K*, *Figure 6—figure supplement 1*). The PRCs are arranged adjacent to each other with the first PRC (PRC1) positioned medially, the second PRC (PRC2) in the middle, and the third PRC (PRC3) laterally. The dorsal rhabdomeric and ciliary eyespots are composed of one PRC and one PC.

RNA in situ hybridization revealed the specific expression of Mfu-xenopsin in both dorsal and ventral rhabdomeric eyes but not in the ciliary eyes. Further, double FISH with Mfu-r-opsin3 (expressed in all rhabdomeric PRCs) confirmed Mfu-xenopsin expression in all three PRCs of the ventral eye (*Figure 6E–E'''*). In dorsal rhabdomeric eyespot, however, in addition to its expression in the Mfu-r-opsin3+ PRC, we detected Mfu-xenopsin in an adjacent cell (*Figure 6F–F'''*).

To assess the presence of ciliary structures in the ventral and dorsal eye and to achieve quantitative data on the surface extension of microvillar and ciliary structures, we analyzed a 3D electron microscopic data set of ventral and dorsal eyes in a 72 hpf stage larva in detail. In the lateral and the medial cells of the ventral eyes, only a basal body with an accessory centriole underneath the apical

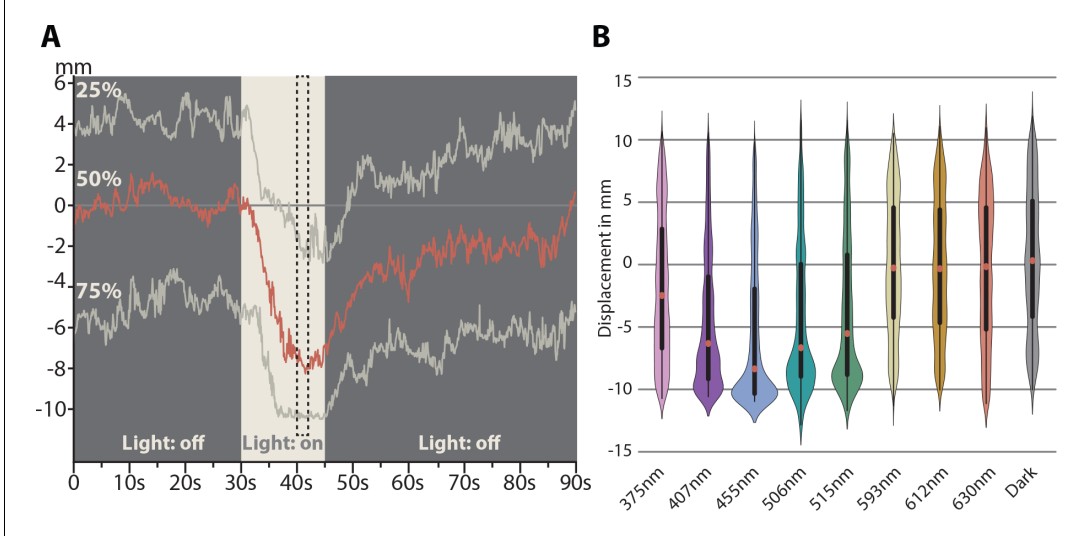

**Figure 5.** Spectral response of *Tricellaria inopinata* larvae. (**A**) One-dimensional displacement of larvae during stimulation with blue (454 nm) light. Each recording started with no stimulus for 30 s. Afterwards, the light stimulus was activated for 15 s, followed by another 45 s in darkness. To generate violin plots, all tracked positions during a time of guaranteed illumination were used (seconds 40 to 42, dashed box). (**B**) Violin plot of the spectral response of the larvae. The animals show the greatest displacement under blue light (454 nm). Within the green and violet spectrum, the animals still respond positively, but further in the ultraviolet and wavelength beyond yellow (593 nm) only weak reactions were detectable. Violin plots based on videos containing between 50 to 230 animals each: 375 nm n = 5; 407 nm n = 4; 455 nm n = 5; 506 nm n = 8; 515 nm n = 13; 593 nm n = 4; 612 nm n = 3; 630 nm n = 3; Dark n = 3. See *Figure 5—figure supplement 1* for violin plots of each individual experiment, *Figure 5—source data 1* for raw data of graph in A and *Figure 5—source data 2* for raw data for the graph in B.

The online version of this article includes the following source data and figure supplement(s) for figure 5:

**Source data 1.** Raw data of behavioral experiment on larval displacement during stimulation with blue light.
**Source data 2.** Raw data of behavioral experiments on the spectral response of the larva.
**Figure supplement 1.** Spectral response of the larvae.

cell membrane (in the medial cell we could see an accessory centriole only on the right body side), but no cilia are present, which gives rise to the microvillar brushes (*Figure 6—figure supplement 1*). In contrast, the middle cell bears a long cilium projecting together with the microvillar brushes into the eye cavity (*Figure 6G–K*). This cilium is also visible in light microscopic stainings (*Figure 6L*). We estimated the microvillar surface of the middle PRC as 296 μm$^2$ based on the average diameter of the microvilli, the number of microvilli per area, and the total volume of the space filled by the microvilli assessed from the 3D image stack. The ciliary surface is 10.7 μm$^2$ based on the length and the diameter of the cilium. Accordingly, the ratio of ciliary to microvillar membrane surface is 1:27.7. The PRC of the dorsal eye likewise possesses a long cilium projecting together with the PRC microvilli into the eye cavity (*Figure 6M–N*).

## Discussion

### *M. fuliginosus* is the first organism known to have both xenopsin and c-opsin

Based on phylogenies with broad taxon sampling across the animal kingdom, *Ramirez et al., 2016* and *Vöcking et al., 2017* reported a secondary loss of xenopsins, as well as c-opsins in several major animal groups. Notably, xenopsins and c-opsins were not known to occur together. Annelids are the only group in which both opsin types were found, while other spiralians have only xenopsin and arthropods and deuterostomes have only c-opsins (*Ramirez et al., 2016*; *Rawlinson et al., 2019*; *Vöcking et al., 2017*). But even within annelids, mutually exclusive distribution of these opsins was reported. Xenopsin was only found in the basally branching oweniids (*Vöcking et al., 2017*), whereas c-opsins were only found in *Platynereis dumerilii* (*Arendt et al., 2004*) and sabellids (*Bok et al., 2017*) and genomic loss of both opsins are evident for *Capitella teleta* and *Helobdella*

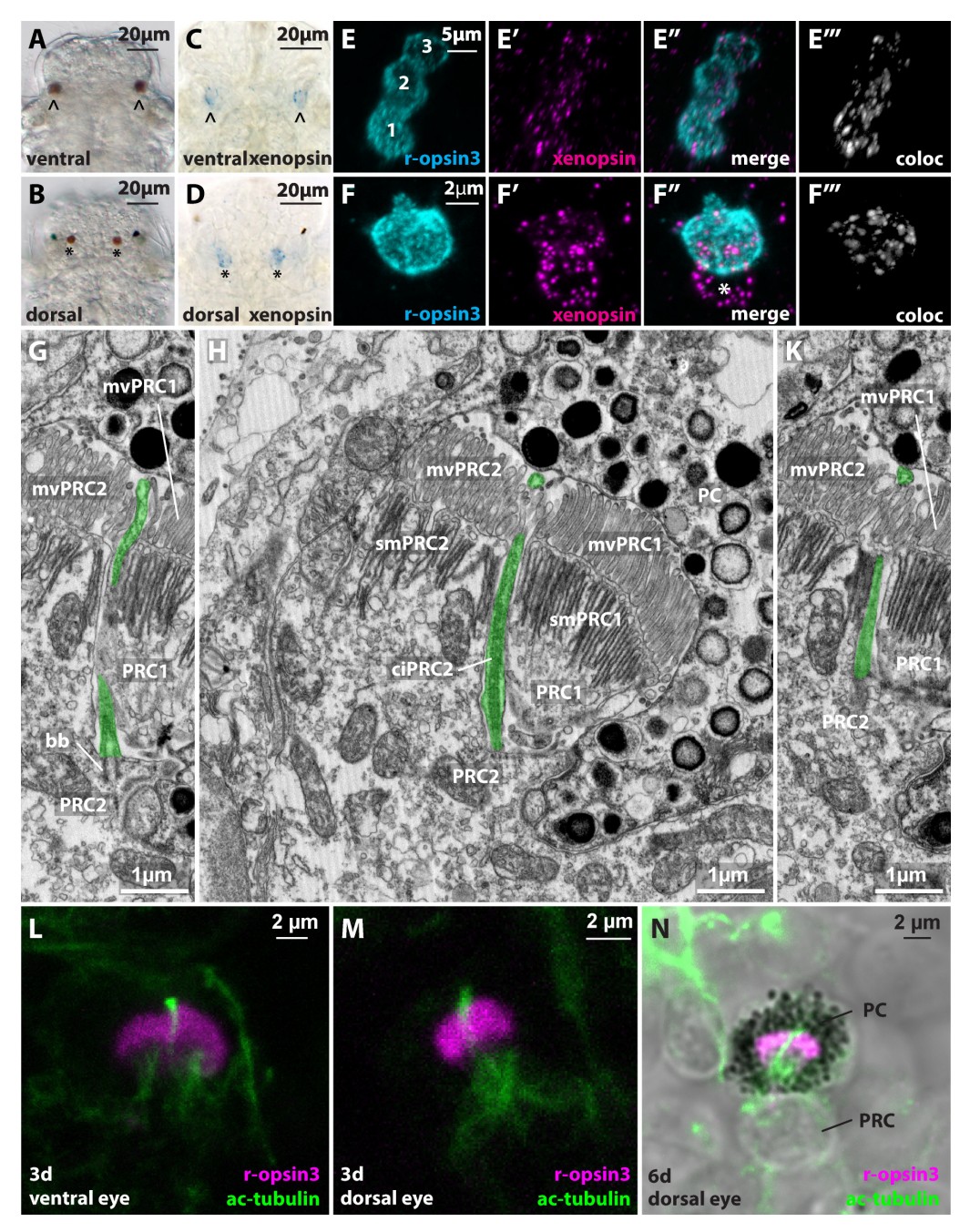

**Figure 6.** Xenopsin in the dorsal and ventral eyes of *Malacoceros fuliginosus*. (**A,B**) Light micrographs of ventral (arrowhead) and dorsal (asterisk) microvillar eyes at 48 hpf. (**C,D**) WMISH of Mfu-xenopsin in the ventral (arrowhead) and dorsal (black asterisk) eyes. (**E–F'''**) Double FISH of Mfu-xenopsin and Mfu-r-opsin3. Mfu-xenopsin co-localizes with Mfu-r-opsin3 in all three PRCs of the ventral eye (**E''**). Numbers indicate the PRCs in the order of their development. (**F–F'''**) Mfu-xenopsin and Mfu-r-opsin3 colocalization in the dorsal eye PRC (**F'''**). Mfu-xenopsin is also expressed in an adjacent cell (white asterisk). (**G–K**) Ultrastructure of the second ventral eye photoreceptor cell (PRC2) depicting the cilium (ciPRC2, highlighted in green). (**L–N**) Antibody labeling against Mfu-r-opsin3 and acetylated alpha-tubulin reveals a prominent cilium emerging in between the r-opsin3+ microvilli in both the ventral and the dorsal eye. bb: basal body; mvPRC1: microvilli of PRC1; mvPRC2: microvilli of PRC2; PRC1: first PRC; smPRC1: submicrovillar cisternae of PRC1; smPRC2: submicrovillar cisternae of PRC2.

The online version of this article includes the following figure supplement(s) for figure 6:

**Figure supplement 1.** The first (PRC1) and third (PRC3) photoreceptor cell of the ventral eye of *Malacoceros fuliginosus* bear no cilia, but exhibit basal bodies (bb) close to the apical surface.

*robusta* (*Vöcking et al., 2017*). To our knowledge, not a single organism has been hitherto reported to have both a c-opsin and a xenopsin. No evolutionary or functional explanation has been given so far as to why xenopsins and c-opsins do not co-occur in any of the animals screened. Accordingly, some uncertainty remained, whether the distinction between xenopsins and c-opsins is a mere tree inference artefact. We now provide evidence for an independent origin of these opsins based on thorough molecular phylogeny, the exon-intron structure of the genes and a clear case of co-occurrence in a single species, *M. fuliginosus*.

## Xenopsin is employed in the eyes of several protostomes

In vertebrates, c-opsins constitute the visual pigments of the retinal rods and cones in the eyes and serve additional functions when expressed in the pineal or deep brain PRCs (*Blackshaw and Snyder, 1999*; *Kawano-Yamashita et al., 2014*; *Hankins et al., 2014*). In protostomes, c-opsins were only reported from annelids and arthropods (*Bok et al., 2017*; *Cronin and Porter, 2014*; *Hering and Mayer, 2014*; *Ramirez et al., 2016*; *Vöcking et al., 2017*). Moreover, the expression of c-opsin has not been reported from PRCs in cerebral eyes, but from extraocular brain photoreceptors and in the case of the annelid subgroup of Sabellida in tentacular crown eyes (*Arendt et al., 2004*; *Beckmann et al., 2015*; *Bok et al., 2017*; *Velarde et al., 2005*; *Verasztó et al., 2018*). Instead, in many protostomes, r-opsins sense light in microvillar eye photoreceptor cells (*Fain et al., 2010*; *Ramirez et al., 2016*; *Terakita, 2005*). Besides, the evidence is increasing that xenopsins play important roles in protostome eyes. So far, xenopsin expression has been reported in the eye PRCs and serially homologous extraocular photoreceptor cells in chiton larva (*Vöcking et al., 2017*), in the eyes of larval brachiopods (*Passamaneck et al., 2011*) and recently in flatworm brain PRCs (*Rawlinson et al., 2019*). In this study, we report expression in the eyes of a larval annelid and eyes of a larval bryozoan. Seemingly, xenopsin is more common in the eyes of those protostomes than hithero anticipated.

## Xenopsin enters cilia

Subcellular targeting of opsins is an important prerequisite for visual perception in PRCs. Being transmembrane proteins, opsins travel integrated into vesicle membranes from the Golgi to the plasma membrane. Once there, they can enter plane plasma membrane areas similar to melanopsin in intrinsic light-sensitive retinal ganglion cells in the vertebrate retina (*Belenky et al., 2003*) and like many of the vertebrate non-visual c-opsins expressed in deep brain receptors, inner layers of the retina and in several other tissues (*Foster and Bellingham, 2004*; *Hunt et al., 2014*). Access to specialized membrane extensions like cilia or microvilli depends on specific active transport mechanisms (*Schopf and Huber, 2017*; *Wang and Deretic, 2014*; *Wingfield et al., 2018*). Though the evolution of sequence motifs relevant for protein binding to the respective transport machinery is not well understood, the capability of certain opsin types to enter either cilia or microvilli is seemingly very well conserved. To our knowledge, no c-opsin entering microvilli and no r-opsin entering cilia are known. We provide unambiguous evidence that xenopsin enters cilia in *T. inopinata*. Likely, this is also the case in the eye photoreceptor cells of the larval brachiopod *Terebratalia*, where xenopsin expressing cells show a ciliary organization (*Passamaneck et al., 2011*) and in brain PRCs in the flatworm *Maritigrella* (*Rawlinson et al., 2019*). The subcellular localization of xenopsin in PRCs of larval *Leptochiton asellus* (*Vöcking et al., 2017*) and larval *M. fuliginosus* is not clear since custom-made antibodies against the opsin did not reveal positive staining. However, in *Leptochiton asellus*, the presence of prominent ciliary structures beside microvilli expressing r-opsin provides the structural prerequisites for a similar opsin targeting. Similarly, in *M. fuliginosus* the dorsal eye PRC and the second ventral eye PRC bear a prominent cilium in addition to microvilli. Whether the xenopsin expressed enters these cilia remains speculative. Only for xenopsin A exist data on the subcellular localization of the protein (always in cilia), but so far not for xenopsin B. The first and third PRC of the ventral eye bear in addition to microvilli only basal bodies and accessory centriols close to the apical surface, which may be remnants of cilia. The xenopsin expressed in these cells may not enter any surface extensions, enter microvilli or may even not be translated into protein. If it would enter microvilli in certain PRCs of *M. fuliginosus*, xenopsin would be the first opsin group known to have the capability to enter both cilia and microvilli.

## Evolution of bilaterian eye PRCs

For a long time, hypotheses on the evolution of eye PRCs focused mainly on PRCs expressing either r-opsin or c-opsin. R-opsin expressing cells employing the same kind of phototransduction cascade and with similar electrophysiological responses are found in the eyes of protostomes and deuterostomes (*Arendt et al., 2002*; *Gomez et al., 2009*; *Fain et al., 2010*; *Koyanagi et al., 2005*; *Panda et al., 2002*; *Shichida and Matsuyama, 2009*). Accordingly and due to conserved patterns in development, the presence of these PRCs already in the eyes of the bilaterian ancestor has been suggested (*Arendt, 2003*; *Arendt, 2008*; *Arendt et al., 2004*; *Fernald, 2006*; *Gehring, 2014*; *Lamb, 2013*; *Shubin et al., 2009*). Though c-opsins detect light in rods and cones of the vertebrate retina, its ancestral expression is assumed in brain extraocular photoreceptors (*Arendt, 2008*; *Arendt et al., 2004*; *Shubin et al., 2009*). Only in such cells were c-opsins found in arthropods (*Beckmann et al., 2015*; *Velarde et al., 2005*) and annelids (*Arendt et al., 2004*). Similarly, many kinds of non-visual c-opsins of vertebrates like encephalopsins, TMT-opsins, or VA-opsins are found in the brain in addition to possible functions in the inner layers of the retina (*Hunt et al., 2014*; *Pérez et al., 2019*). Accordingly, the employment of c-opsin cells in the visual cells of cerebral eyes evolved later, most probably in the lineage leading to chordates (*Vopalensky et al., 2012*).

Morphological data on the presence of ciliary PRCs in the eyes of several less studied protostome animals like bryozoans (*Reed et al., 1988*; *Woollacott and Eakin, 1973*; *Woollacott and Zimmer, 1972*), gastrotrichs (*Woollacott and Eakin, 1973*) and nemerteans (*von Döhren and Bartolomaeus, 2018*) are, however, in conflict with this scenario. *Vöcking et al., 2017* proposed that in addition to rhabdomeric and c-opsin, xenopsin is a third important player in PRC and eye evolution questioning a common origin of ciliary eye PRCs in protostomes and deuterostomes. Our study is providing further support for this view.

Though in protostomes, c-opsins only exist in annelids and arthropods, ancestral employment in extraocular brain PRCs is still likely. But seemingly many kinds of protostome ciliary PRCs do not employ c-opsin. Instead, cellular expression of xenopsin is reported from ciliary PRCs in the eyes of larval brachiopods (*Passamaneck et al., 2011*) and bryozoans (this study) and meanwhile also from extraocular and eye ciliary PRCs in flatworms (*Rawlinson et al., 2019*). Further, the presence of xenopsin is also known from molluscs, annelids, chaetognaths, and rotifers (*Ramirez et al., 2016*; *Rawlinson et al., 2019*; *Vöcking et al., 2017*). Coexpression with r-opsin is evident in a larval chiton (*Vöcking et al., 2017*) and an annelid (this study) in microvillar eye PRCs, which partly also exhibit ciliary structures. Xenopsin may have been co-opted by these mainly microvillar cells (*Figure 7* scenario A), but it is also conceivable that the observed cellular coexpression with r-opsin in two subgroups of lophotrochozoans points towards an evolutionary link between ciliary and microvillar PRCs. During the evolution of protostomes eyes, formerly microvillar PRCs may have changed into mixed microvillar/ciliary cells coexpressing r-opsin and xenopsin (*Figure 7*, scenario B). Since co-expression is evident from xenopsin A and B, this happened likely before the diversification of xenopsins in protostomes. Clear hypotheses on the ancestral targeting of xenopsin (cilia and/or microvilli) may need further investigation, but existing data so far point towards cilia as targets. Even the presence of mixed cells in the eyes of the last common ancestor of bilaterians is conceivable (*Figure 7*, scenario C), since opsin tree topology suggests a genomic loss of xenopsin in the deuterostome stem lineage. Within protostomes, the mixed organization was retained in some extant organisms (several molluscs, certain annelids) and transformed in other organisms into a purely ciliary or microvillar organization going along with loss or downregulation of r-opsins (bryozoans, brachiopods) or xenopsins (arthropods, certain annelids), respectively. Such a hypothesis also raises the question, whether the co-occurrence of ciliary and microvillar PRCs within the same eye as known, for example from several molluscs (*Bartolomaeus, 1992*; *Blumer, 1998*; *Salvini-Plawen, 2008*) or the larva of the polyclad flatworms (*Eakin and Brandenburger, 1981*; *Rawlinson et al., 2019*) may be the result of integrating or co-opting ciliary cells into microvillar eyes or caused by duplication and diversification of formerly mixed microvillar/ciliary PRCs. Interestingly, the polyclad flatworm *Maritigrella crozieri* has several kinds of ciliary PRCs, which are expressing xenopsin in adults outside of pigmented eyes, in larval epidermal eyes, and in larval cerebral eyes, which also contain r-opsin expressing microvillar PRCs (*Rawlinson et al., 2019*). The evolutionary origin of the extraocular PRCs and the epidermal eyes is unclear. Nonetheless, developmental data from *Schmidtea mediterranea*, indicate homology of flatworm cerebral eyes to those of other protostomes (*Lapan and Reddien, 2011*; *Lapan and*

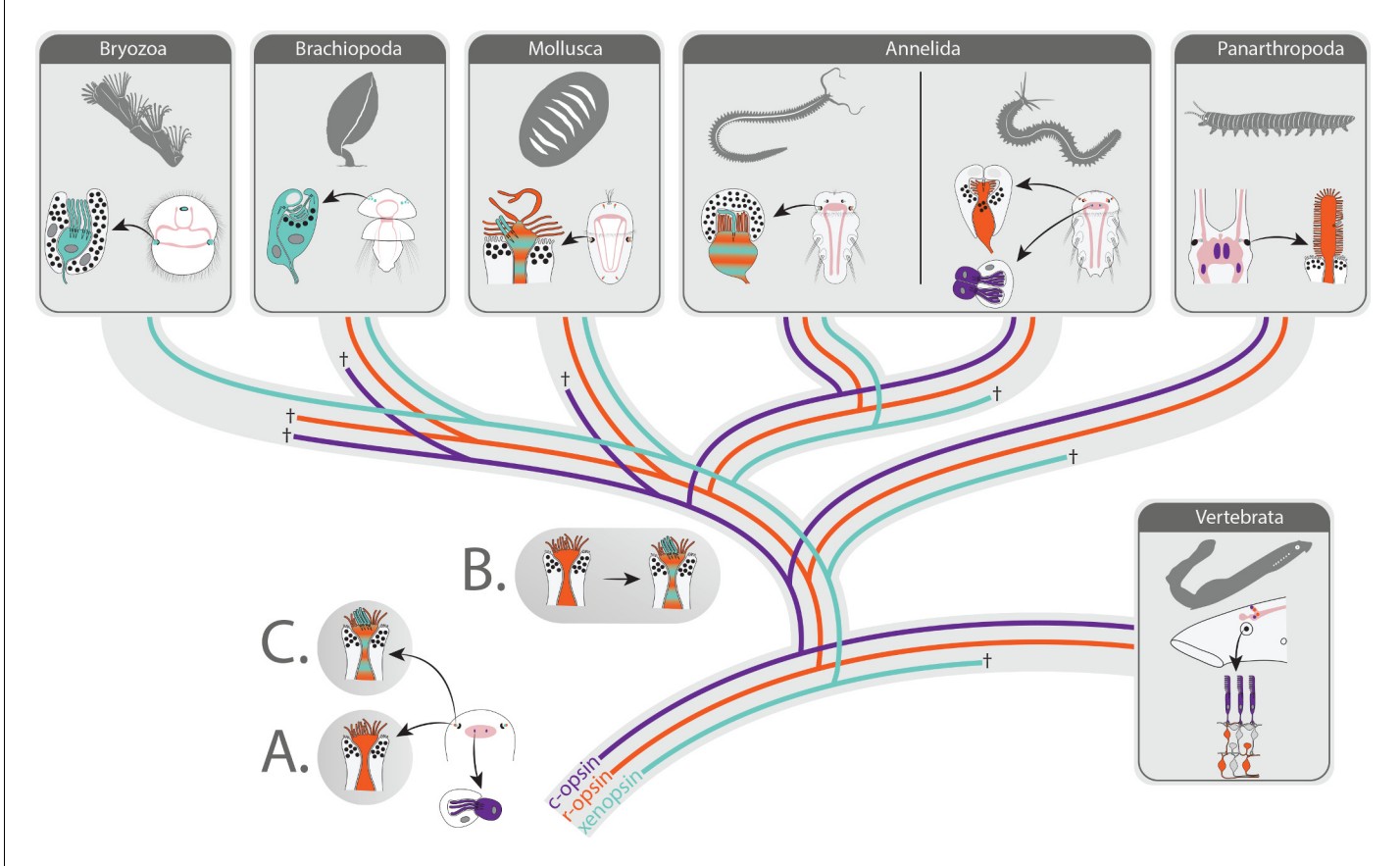

**Figure 7.** Scenarios on eye PRC evolution in Bilateria. The bilaterian ancestor had extraocular c-opsin+ ciliary PRCs. These became integrated into the eyes in the lineage leading to vertebrates and were lost in many protostomes along with secondary loss of c-opsin. Scenario A: Cerebral eyes contained microvillar r-opsin+ PRCs in the bilaterian ancestor. Xenopsin was co-opted several times independently by microvillar PRCs, and eye PRCs were several times independently transformed into or replaced by ciliary xenopsin+ PRCs. Scenario B: Ancestral r-opsin+ microvillar eye PRCs were transformed once into mixed microvillar/ciliary PRCs coexpressing r-opsin and xenopsin. In extant organisms, those were retained or changed into purely microvillar r-opsin+ or ciliary xenopsin+ PRCs going along with genomic loss or downregulation of xenopsin or r-opsin, respectively. Scenario C: Mixed ciliary/microvillar PRCs were already present in the bilaterian ancestor.

*Reddien, 2012*). Therefore, origination from eyes with mixed r-opsin/xenopsin+ microvillar/ciliary PRCs is conceivable.

## Xenopsin function and physiology

Strong phototactic responses, as we observed in the larva of *T. inopinata*, depend on directional detection of light by pigmented eyes. Since we could not find any opsin other than xenopsin expressed in the eyes of *T. inopinata*, we suggest that the xenopsin here is responsible for light reception and triggers the phototactic behavior of the larva. Accordingly, it has a similar visual function as r-opsins have in microvillar eye PRCs of several other protostomes (*Fain et al., 2010*; *Jékely et al., 2008*; *Neal et al., 2019*; *Randel et al., 2013*). The ciliary surface of the PRC in *T. inopinata* is even three times larger than the microvillar surface of the middle eye PRC in *M. fuliginosus*. Notably, heterologous expression of *Maritigrella crozieri* xenopsin suggests that it acts mainly via $G\alpha_i$ and possibly to a lower extent also via $G\alpha_s$, but not via $G\alpha_q$ signaling (*Rawlinson et al., 2019*) as r-opsins do (*Fain et al., 2010*; *Shichida and Matsuyama, 2009*). Hence, similar to the case of r-opsins and c-opsins, downstream signaling and even the cellular electrophysiological response may also be different upon the excitation of r-opsins and xenopsins.

Most PRCs express only one type of visual opsin. If PRCs employ more than one opsin, they are usually from the same subgroup of visual pigments and activate the same transduction cascade (*Applebury et al., 2000*; *Arikawa et al., 2003*; *Dalton et al., 2015*; *Isayama et al., 2014*;

*Katti et al., 2010*; *Parry and Bowmaker, 2016*; *Rajkumar et al., 2010*), which is suggested, ultimately, to expand the visual spectrum. The same function is assumed in the annelid *P. dumerilii*, where Go-opsin and r-opsin co-occur in eye PRCs (*Gühmann et al., 2015*). If Gα$_i$ signaling of xenopsin is conserved in protostomes, PRCs coexpressing xenopsin and r-opsin, as we observed in *M. fuliginosus* and are known in *Leptochiton asellus* (*Vöcking et al., 2017*), might potentially be polymodal sensory cells with complex physiology capable of integrating different stimuli by activation of different signaling cascades. The ciliary surface of the middle eye PRC of *M. fuliginosus* is nearly 30 times less than the microvillar surface. This may hint to a minor role of xenopsin signaling in this eye, but other parameters like the efficiency of the specific sensory transduction pathway and the protein content in the membrane certainly impact the sensitivity as well. The contribution of cilia in light detection may be higher in eyes where microvilli are accompanied by higher numbers of cilia as described in several molluscs (*Blumer, 1995*; *Blumer, 1996*; *Hughes, 1970*; *Zhukov et al., 2006*).

## Conclusion

Xenopsin seems to be an important visual pigment in protostome eyes. This opsin type was overlooked for a long time, probably because most molecular data on protostome light perception are from arthropods, which secondarily lost xenopsin. *M. fuliginosus* is the first organism known to have xenopsin and c-opsin corroborating the distinct evolutionary origin of these opsin types inferred from phylogenetic and gene structure analysis. All other organisms studied thus far, for reasons unknown, have either c-opsin or xenopsin. Xenopsin, like c-opsin, enters cilia. In protostomes, it is employed in purely ciliary PRCs and found coexpressed with r-opsin in microvillar PRCs that also have ciliary structures. Xenopsin or xenopsin expressing cells might have been recruited several times independently in the eyes of protostomes. Alternatively, these eyes already early in evolution employed a possibly polymodal r-opsin+ and xenopsin+ microvillar/ciliary PRCs. Further studies on the employment of xenopsin in protostomes will be of high interest for a better understanding of evolution, function, and plasticity of animal photoreceptor cells and eyes. Counterparts of vertebrate c-opsin employing ciliary PRCs in protostomes probably exist only in certain annelids and arthropods, since c-opsin according to available sequence resources has been lost in all other protostome animals.

# Materials and methods

**Key resources table**

| Reagent type (species) or resource | Designation | Source or reference | Identifiers | Additional information |
|---|---|---|---|---|
| Gene (*Tricellaria inopinata*) | Xenopsin | Genbank | MT901641 | |
| Gene (*Malacoceros fuliginosus*) | Xenopsin | Genbank | MT901639 | |
| Gene (*Malacoceros fuliginosus*) | Ciliary opsin | Genbank | MT901640 | |
| Strain, strain background (*Malacoceros fuliginosus*) | Wild type | University of Bergen, Sars Centre for Marine Molecular Biology | NCBITaxon: 271776 | |
| Antibody | mouse monoclonal anti-acetylated α-tubulin IgG1 | Sigma-Aldrich | RRID:AB_609894 | Dilution 1:300 (Mfu) 1:50 (Tin) |
| Antibody | Rat polyclonal anti-Mfu-r-opsin3 IgG | University of Bergen, Sars Centre for Marine Molecular Biology | | 1:100 |
| Antibody | Rabit polyclonal anti-Tin-xenopsin IgG | University of Bergen, Sars Centre for Marine Molecular Biology | | 1:500 |

*Continued on next page*

*Continued*

| Reagent type (species) or resource | Designation | Source or reference | Identifiers | Additional information |
|---|---|---|---|---|
| Antibody | Alexa Fluor 633 goat monoclonal anti-rat IgG | ThermoFisher Scientific | RRID:AB_2535749 | 1:500 |
| Antibody | Alexa Fluor 488 goat momoclonal anti-mouse IgG | ThermoFisher Scientific | RRID:AB_2535764 | 1:500 |
| Recombinant DNA reagent | PGem-T-Tin-xenops (plasmid) | University of Bergen, Sars Centre for Marine Molecular Biology | | Used for synthesizing WMISH probes |
| Recombinant DNA reagent | PGem-T-Mfu-xenops (plasmid) | University of Bergen, Sars Centre for Marine Molecular Biology | | Used for synthesizing WMISH probes |
| Recombinant DNA reagent | PGem-T-Mfu-cops (plasmid) | University of Bergen, Sars Centre for Marine Molecular Biology | | Used for synthesizing WMISH probes |
| Sequence-based reagent | Mfu-xenops-WMISH forward primer (5'- > 3') | University of Bergen, Sars Centre for Marine Molecular Biology | | 5'-CACCATCATGTTGAATAA TGACTCCTACTC-3' |
| Sequence-based reagent | Mfu-xenops-WMISH reverse primer (5'- > 3') | University of Bergen, Sars Centre for Marine Molecular Biology | | 5'-GATTCGTGGAATGCTG ATTTGTGAC-3' |
| Sequence-based reagent | Mfu-cops-WMISH forward primer (5'- > 3') | University of Bergen, Sars Centre for Marine Molecular Biology | | 5'-ATCACACAGGATATCAC AAATGCCTCAG-3' |
| Sequence-based reagent | Mfu-cops-WMISH reverse primer (5'- > 3') | University of Bergen, Sars Centre for Marine Molecular Biology | | 5'-GCAATAACGATGTCACC TGGACATTG-3' |
| Sequence-based reagent | Tin_xenopsin-WMISH forward primer (5'- > 3') | University of Bergen, Sars Centre for Marine Molecular Biology | | 5'-CTTATGGTCATTGCTGT-3' |
| Sequence-based reagent | Tin_xenopsin-WMISH reverse primer (5'- > 3') | University of Bergen, Sars Centre for Marine Molecular Biology | | 5'-CACCCTGCCATTAGTC-3' |
| Sequence-based reagent | Tin_xenopsin-WMISH forward nested primer (5'- > 3') | University of Bergen, Sars Centre for Marine Molecular Biology | | 5'-TGGGGGTTGTTTTGGTCGT-3' |
| Sequence-based reagent | Tin_xenopsin-WMISH reverse nested primer (5'- > 3') | University of Bergen, Sars Centre for Marine Molecular Biology | | 5'-CTGTTGCCTTCTTCTCTCGT-3' |
| Commercial assay or kit | Superscript III First-Strand Synthesis System | ThermoFisher Scientific | Catalog number: 18080051 | |
| Commercial assay or kit | RNeasy Mini Kit | Qiagen | Catalog number: 74104 | |
| Software, algorithm | IQ-TREE | http://www.iqtree.org/ | RRID:SCR_017254 | |
| Software, algorithm | Phylobayes-MPI | https://github.com/bayesiancook/pbmpi | RRID:SCR_006402 | |
| Software, algorithm | MAFFT 7 | https://mafft.cbrc.jp/alignment/server/ | RRID:SCR_011811 | |
| Software, algorithm | CLC Main Workstation | Qiagen | RRID:SCR_000354 | |
| Software, algorithm | ImageJ | NIH | RRID:SCR_003070 | |
| Software, algorithm | Imaris 8.41 | Bitplane | RRID:SCR_007370 | |

## Animal culture

Adults of the polychaete *Malacoceros fuliginosus* (*Claparède, 1868*) were collected from Pointe de Mousterlin, Fouesnant, France. The animals were maintained in the lab facility in sediment containing seawater tanks at 18°C and fed with ground fish food flakes (TetraMin granules, Tetra). Mature males and females were picked, rinsed several times with filtered seawater, and kept in separate bowls until they spawned. Staging was started from the time gametes were combined in a fresh bowl. Bowls were kept at 18°C under 12:12 hr light-dark cycle, and water was replaced every day or every second day. Larvae were fed with the microalga *Chaetoceros calcitrans* from 24 hpf onwards after each water change. Colonies of the bryozoan *Tricellaria inopinata* (d'Hondt & Occhipinti Ambrogi, 1985) were collected in Brest, France (48°23'38.3"N 4°25'57.4"W). The colonies were maintained at 18°C under 12:12 hr light-dark cycle in the lab animal facility.

## RNA-seq and transcriptome assembly

For studies on *M. fuliginosus,* we used transcriptomic resources prepared in an earlier study (*Kumar et al., 2020*) from pooled larvae of several stages. For *T. inopinata,* we performed RNA-seq and de novo transcriptome assembly. The release of larvae was triggered by the onset of light in the tanks. Two hours later, swimming larvae were attracted by a light bulb and cryo-fixed. We extracted total RNA using the Agencourt RNAdvance Tissue Kit (Beckman Coulter, Brea, California). Library preparation and sequencing were performed by EMBL (Heidelberg, Germany) Genomics Core Facility using cation-based chemical fragmentation of RNA, Illumina (San Diego, California) Truseq RNA-Sample Preparation Kit and 1 lane of 100 bp paired-end read sequencing on Illumina HiSeq 2000. We used Cutadapt 1.2.1 (RRID:SCR_011841) for trimming and the ErrorCorrectReads tool implemented in Allpaths-LG (RRID:SCR_010742) for error correction of the raw reads and Trinity (RRID: SCR_013048) for de novo assembly. We performed two rounds of RNA-seq and assembly. For the first data set (assembly 1) the collected colonies were thoroughly cleaned. However, by microscopic inspection, we observed a minor proportion of zooids of other bryozoan species, which were dispersed across the colonies and could not be entirely removed. We assessed contamination of the respective assembly by screening for cytochrome oxidase subunit I (COI), and we found sequences indeed from four different bryozoans, but none from other animal groups. The second data set (assembly 2) showed contamination also with sequences from other taxa and was only used for corroboration and elongation of sequences retrieved from assembly 1.

## Opsin tree inference

Retrieved opsin sequences from *T. inopinata* and *M. fuliginosus* were added to the set of opsin sequences (https://doi.org/10.7554/eLife.23435.009) analyzed by *Vöcking et al., 2017*. Sequences were first aligned with MAFFT 7 (RRID:SCR_011811) with option E-INS-I, ambiguously aligned N- and C-terminal regions were trimmed, sequences shorter than 160 amino acids removed and the remaining set of sequences again aligned with MAFFT seven with option E-INS-I. The output was manually edited to remove gap rich regions and ambiguously aligned positions. Maximum-likelihood phylogenetic analyses were run with IQ-TREE 1.5.5 (RRID:SCR_017254) with model LG+F+R8 chosen by ModelFinder, SH-like approximate likelihood ratio test (1000 replicates), ultrafast bootstrap (1000 replicates) and approximate Bayes test for estimating branch support, unsuccessful iterations to stop tree searching set to 500 and perturbation strength to 0.2. Bayesian analysis was performed with Phylobayes-MPI 1.6 (RRID:SCR_006402) running three chains for 90.000 cycles using the dataset specific substitution matrix (DS-GTR) generated by *Vöcking et al., 2017* with parametric $\Gamma$ modeling. Phylogenetic convergence of the chains was assessed with bpcomp.

## Opsin gene structure analysis

For the two opsins found in *M. fuliginosus* and the one found in *T. inopinata,* the whole genes were cloned from genomic DNA for subsequent analysis of exon-intron boundaries. Genomic DNA was extracted with the Nucleospin Tissue Kit (Machery-Nagel, Düren, Germany) and tested for fragment length larger than 20 kb. As a starting point, gene-specific primers were designed based on the transcript sequences. For genome walking, four libraries were prepared with Universal Genome Walker Kit (Takara Bio, Mountain View, California) by enzymatic digestion and used for sequence elongation starting from exonic fragments. In parallel, long amplicons bridging smaller introns were also directly

amplified from genomic DNA using LA Taq (Takara Bio), iProof (Biorad, Hercules, California, USA), and HotStarTaq Plus (Qiagen, Hilden, Germany) polymerases. Obtained amplicons up to 8 kb were cloned using pGEM-T easy Vector (Promega, Madison, Wisconsin) TOPO XL PCR cloning kit (Thermo Fisher Scientific), TopTen chemically competent cells (Thermo Fisher Scientific, Waltham, Massachusetts) and Sanger sequenced. Obtained sequences were used to design further primers for ongoing sequence elongation. Read assembly was performed with CLC Main Workstation (RRID:SCR_000354) 7.1. Gene structures of the opsins from *M. fuliginosus* and *T. inopinata* were determined based on the cloned genomic and the protein sequences retrieved from RNA-seq using WebScipio (*Hatje et al., 2011*). The obtained gene structures were together with the gene structures prepared by *Vöcking et al., 2017* mapped onto the un-curated sequence alignment using Genepainter (*Hammesfahr et al., 2013*) to identify conservation of intron positions and phases.

## Custom antibodies

Custom polyclonal antibodies were prepared and affinity-purified against peptides of Tin-xenopsin by 21$^{st}$ Century Biochemicals (Marlboro, Massachusetts) and Mfu-r-opsin-3 by Eurogentec (Liège, Belgium). For Tin-xenopsin, the peptide sequences VKAAGKKFGGDDAASQ from the 3$^{rd}$ cytoplasmic loop and ATKPAPSATQAPREKKATAL from the cytosolic tail and for Mfu-ropsin3 RHSE VPSGDGKKDTL and CKNRAIDKGGDEESDN both from the cytoplasmic tail were chosen as antigens. To assure antigen specificity, all peptide sequences were blasted against the *T. inopinata,* and *M. fuliginosus* transcriptome and gave only the respective opsin sequences as hits. The antibodies raised against the peptides ATKPAPSATQAPREKKATAL and RHSEVPSGDGKKDTL gave the best results and were used for the stainings.

## Immunohistochemistry

*M. fuliginosus* larvae were first relaxed with 1:1 MgCl2-seawater for 3–5 min before fixing them in 4% PFA (in 1X PBS, 0.1% Tween20) for 30 min at RT. After fixing, the samples were washed two times in PTW followed by two washes in THT (0.1 M Tris pH 8.5, 0.1% Tween20). Blocking was in 5% sheep serum in THT for 1 hr before incubating in primary antibodies rat anti-Mfu r-opsin3, 1:100; monoclonal mouse anti-acetylated α-tubulin IgG (RRID:AB_609894) from Sigma Aldrich (Saint Louis, Missouri) 1:300 (Mfu), 1:50 (Tin), anti-Tin-xenopsin, 1: 500) for 48 hr at 4°C. The samples were then subjected to two 10 min washes in 1 M NaCl in THT followed by five 30 min washes in THT before incubating in secondary antibodies (Alexa Fluor 633 goat anti-rat IgG (RRID:AB_2535749), 1:500 and Alexa Fluor 488 goat anti-mouse IgG (RRID:AB_2535764), 1:500 (Thermo Fisher Scientific) overnight at 4°C. Next, the samples were washed in THT, two 5 min washes followed by five 30 min washes. Specimens were stored in embedding medium (90% glycerol, 1x PBS and 0.25% DABCO) at 4°C.

## In situ hybridization

For gene cloning cDNA, was prepared from total RNA with Super Script II (Thermo Fisher Scientific), sequences of interest were PCR amplified with gene-specific primers, and amplicons were subsequently ligated into pgemT-easy vector (Thermo Fisher Scientific) and cloned into Top10 chemically competent *E. coli* (Thermo Fisher Scientific). Sanger sequencing was used to verify the cloned sequences before DIG- and FITC-labeled sense and antisense probes were generated with T7 and SP6-RNA Polymerases (Roche, Basel, Switzerland) or with Megascript Kit (Thermo Fisher Scientific). If needed, Smarter Race (Takara Bio) was used to elongate ends of transcript sequences. In situ hybridization experiments were performed as described previously (*Vöcking et al., 2015*) if formamide based hybridization buffers were used. Otherwise, we followed # (*Sinigaglia et al., 2018*) # for urea-based hybridization buffers. In brief, animals were fixed in 4% PFA in phosphate buffer and with Tween20 (PTW; pH 7.4) and subsequently washed and stored in methanol. For *Tricellaria inopinata* larvae, a 2 min prefixation with 0.3% Glutaraldehyde in 4% PFA was necessary. After rehydration in PTW, samples were briefly digested with Proteinase K, washed and prehybridized in hybridization with or without 5% dextran. Samples were hybridized with RNA probes for 72 hr. *Tricellaria inopinata* larvae required that each washing step after hybridization was extended to 30 min. Stainings were done with a combination of FastBlue (Sigma-Aldrich) and Fast Red (Roche). The significance of expression signals was evaluated by using sense probes as control experiments. All

in situ hybridization experiments were performed on at least 15 specimens per gene for each sense, and anti-sense probe and the experiments were repeated at least two times.

## Light microscopy

Light microscopic images were taken using Eclipse E800 (Nikon, Tokyo, Japan) and Examiner A.1 (Zeiss, Oberkochen, Germany), and confocal images were taken with an SP5 confocal microscope (Leica, Wetzlar, Germany). Image stacks were processed with ImageJ (RRID:SCR_003070), Imaris (RRID:SCR_007370) and Adobe Photoshop CC (RRID:SCR_014199).

## Electron microscopy

For electron microscopic studies, two kinds of sample preparations were used. For chemical fixation larvae were relaxed for 3 min in 7% $MgCl_2$ and seawater mixed 1:1 and then fixed in 2.5% glutaraldehyde in PBS, postfixed in 1% Osmium tetroxide in the same buffer, en-bloc stained with reduced Osmium, dehydrated in a graded ethanol series and embedded in Epon/Araldite as described in *Vöcking et al., 2015*. Cryo-fixation was performed at the EM Core facility at EMBL (Heidelberg, Germany). Larvae were relaxed as described above and then high-pressure-frozen with hexadecene acting as filler in an HPM 010 from Balzers (Balzers, Liechtenstein). Freeze substitution with 2% $OsO_4$ and 0.1% uranyl acetate in a mixture of 95% acetone and 5% water was performed in a Leica (Wetzlar, Germany) EM AFS2 for 46 hr at $-90°C$. Samples were slowly warmed to $-30°C$, kept at this temperature for 6 hr, and slowly warmed to 0°C before they were taken out from the freeze-substitution device. Samples were rinsed several times in acetone at 0°C and at room temperature, stepwise transferred to Epon, and cured for 48 hr at 60°C.

Serial sections of 70 nm were cut with an ultra 35° diamond knife (Diatome, Biel, Switzerland) on a UC7 ultramicrotome (Leica) and collected on Beryllium-Copper slot grids (Synaptek, Reston, Virginia, USA) coated with Pioloform (Ted Pella, Redding, California, USA) and counterstained with 2% uranyl acetate and lead citrate. Complete series were imaged with STEM-in-SEM as described by *Kuwajima et al., 2013* at a resolution of 4 nm/ pixel with a Supra 55VP (Zeiss, Oberkochen, Germany) equipped with Atlas (Zeiss) for automated large field of view imaging. Acquired images were processed with Adobe Photoshop CC, first registered rigidly followed by affine and elastic alignment (*Saalfeld et al., 2012*) with TrakEM2 (*Cardona et al., 2012*) implemented in Fiji (RRID:SCR_002285).

## Behavioral assays of *T. inopinata* larva

Freshly hatched larvae were placed in a small clear plastic container that was situated inside of a chamber with two infrared long-pass filters (RG610, Reichmann Feinoptic, Brokdorf, Germany) installed above and below the container. The chamber was placed on a Zeiss Stemi 2000 stereo microscope. Through a hole on one side of the chamber, an LED served as the light stimulus. Eight different LEDs covered wavelengths from UV 375 nm to 630 nm in the red part of the spectrum. For each experiment between 50 to 230 freshly hatched larvae were placed inside the chamber. We recorded the reaction of the animals to the stimulus with an industrial monochrome CMOS camera (DMK 23U445, The Imaging Source, Bremen, Germany). Each recording starts in darkness for 30 s, followed by 15 s of illumination and another 45 s of darkness. The response of the animals to each wavelength was assayed at least three times, always with a new batch of animals and an extra four without any light stimulus as a control. From each recording, we removed the background and enhanced the contrast in Fiji. Subsequently, we tracked the animals' position in enhanced recordings for each frame with the Fiji plugin Trackmate (*Tinevez et al., 2017*). The tracking information was used to calculate the mean and median position of the animals for each frame for a single axis. To make different recordings comparable, we used the mean and median position of animals during the initial darkness to subtract from each position for each frame. Boxplots Violinplots were inferred from all the tracked positions of all the animals during a time of guaranteed illumination (second 40 to 42, *Figure 5A*, dashed box).

## Acknowledgements

We are very thankful to Yannick Schwab and Rachel Mellwig at the Electron Microscopy core facility at EMBL Heidelberg for their assistance and advice in cryo-fixation of EM samples and the Arendt

lab at EMBL Heidelberg for hosting cultures of *M. fuliginosus* and *T. inopinata* in their animal facility for EM fixation.

## Additional information

### Funding

| Funder | Grant reference number | Author |
|---|---|---|
| European Commission | FP7-PEOPLE-2012-ITN 317172 (NEPTUNE) | Harald Hausen |

The funders had no role in study design, data collection and interpretation, or the decision to submit the work for publication.

### Author contributions

Clemens Christoph Döring, Suman Kumar, Data curation, Formal analysis, Investigation, Visualization, Methodology, Writing - original draft, Writing - review and editing; Sharat Chandra Tumu, Formal analysis, Investigation, Methodology, Writing - original draft; Ioannis Kourtesis, Formal analysis, Investigation, Methodology; Harald Hausen, Conceptualization, Resources, Data curation, Formal analysis, Supervision, Funding acquisition, Validation, Investigation, Visualization, Methodology, Writing - original draft, Project administration, Writing - review and editing

### Author ORCIDs

Clemens Christoph Döring (iD) https://orcid.org/0000-0002-3545-7882
Suman Kumar (iD) https://orcid.org/0000-0002-2280-9113
Harald Hausen (iD) https://orcid.org/0000-0003-2788-2835

### Decision letter and Author response

Decision letter https://doi.org/10.7554/eLife.55193.sa1
Author response https://doi.org/10.7554/eLife.55193.sa2

## Additional files

### Supplementary files

• Transparent reporting form

### Data availability

Sequencing data have been deposited in Genbank under accession codes MT901639, MT901640, and MT901641. Source data files have been provided for Figures 2 and 5.

The following datasets were generated:

| Author(s) | Year | Dataset title | Dataset URL | Database and Identifier |
|---|---|---|---|---|
| Doering CC, Kumar S, Tumu S, Kourtesis I, Hausen H | 2020 | Malacoceros fuliginosus xenopsin gene, partial cds | https://www.ncbi.nlm.nih.gov/nuccore/MT901639 | NCBI GenBank, MT901639 |
| Doering CC, Kumar S, Tumu S, Kourtesis I, Hausen H | 2020 | Malacoceros fuliginosus ciliary opsin gene, complete cds | https://www.ncbi.nlm.nih.gov/nuccore/MT901640 | NCBI GenBank, MT901640 |
| Doering CC, Kumar S, Tumu S, Kourtesis I, Hausen H | 2020 | Tricellaria inopinata xenopsin gene, partial cds | https://www.ncbi.nlm.nih.gov/nuccore/MT901641 | NCBI GenBank, MT901641 |

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
