## [Decision Letter]

**Acceptance summary:**

Light receptor proteins in eyes come in three major types of which ciliary opsin and rhabdomeric opsin coexist in some animal groups including vertebrates. The newly discovered xenopsin has been reported to be expressed alone or coexist with rhabdomeric opsin. This report describes the first organism with all three opsin categories, the marine annelid *Malacoceros fuliginosus*.

**Decision letter after peer review:**

Thank you for submitting your article "The visual pigment xenopsin is widespread in protostome eyes and impacts the view on eye evolution" for consideration by *eLife*. Your article has been reviewed by two peer reviewers, and the evaluation has been overseen by a Reviewing Editor and Diethard Tautz as the Senior Editor. The reviewers have opted to remain anonymous.

The Reviewing Editor has drafted this decision to help you prepare a revised submission.

Summary:

This is an interesting manuscript as it seems to resolve an important matter in visual opsin evolution by demonstrating presence of the enigmatic xenopsin together with c-opsin in an annelid, leading to the conclusion that xenopsin has evolved in parallel with the other two visual opsins (c-opsins and r-opsins). The study also shows that xenopsin localizes to larval cilia. Thereby it adds to a broader understanding of opsin evolution, eye evolution and the basis of visually guided behavior.

Essential revisions:

Generally, the reviewers found the manuscript well written and data carefully collected and well supported. They specifically commented that ISH and antibody stainings are of high quality and with beautiful transmission electron micrographs.

As an exception, the description of the behavioural assay did not clearly say whether a single experiment (with 50 to 100 animals) was performed, and then each animal's position taken as a data point, or whether the figure builds on multiple (how many?) experiments for each wavelength. Please be aware that the position of 100 animals all released together cannot be treated as independent data points; many if not the majority of such organisms do one or more of the following: avoid each other, follow each other, chemically communicate with each other. Thus, if the figure builds on just one of these experiments per wavelength, the data may not be treated correctly. There is little doubt that there is a reaction, but it would be good (and likely improve the data set) if the experiment was repeated several times with each wavelength, and the mean etc. of these separate experiments was presented.

Figure 2: The evidence for xenopsins and cnidopsins forming a clade still finds low to moderate support only and might therefore represent an artifact. The possibility remains that Xenopsins exist in Lophotrochozoans only and this should be mentioned and visualized in the model.

Even though many nodes in the xenopsin tree are well supported, it seems to suggest the existence of two paralogs in Lophotrochozoa that are distinctly present in molluscs, annelids and platyhelminths. This duplication appears to have occurred at the Lophotrochozoan root. If so, then the ciliary location would only be shown for one representative of one of the paralogs. This is important and should be mentioned.

For Malacoceros xenopsin, the reasoning that xenopsin locates to cilia does not work out because the two rhabdomeric photoreceptors that strongly express it do not have cilia (only basal bodies). Where would the opsin go? There is no evidence for 'rudimentary cilia', they are simply absent (no acetylated tubulin staining). This should be stated as is.

The figures, supplementary figures and figure legends seem to have been mixed in the new version as compared to the previous version. The numbers had disappeared. This confusing mess must be sorted out.

---

## [Author Response]

Essential revisions:Generally, the reviewers found the manuscript well written and data carefully collected and well supported. They specifically commented that ISH and antibody stainings are of high quality and with beautiful transmission electron micrographs.As an exception, the description of the behavioural assay did not clearly say whether a single experiment (with 50 to 100 animals) was performed, and then each animal's position taken as a data point, or whether the figure builds on multiple (how many?) experiments for each wavelength. Please be aware that the position of 100 animals all released together cannot be treated as independent data points; many if not the majority of such organisms do one or more of the following: avoid each other, follow each other, chemically communicate with each other. Thus, if the figure builds on just one of these experiments per wavelength, the data may not be treated correctly. There is little doubt that there is a reaction, but it would be good (and likely improve the data set) if the experiment was repeated several times with each wavelength, and the mean etc. of these separate experiments was presented.

We agree that interactions of the larvae may introduce some bias and thus the new Figure 5 is now based on data from several independent experiments. For each wavelength we performed 3 up to 13 recordings and for each recording a new batch of 50 to 230 larvae was used. For Figure 5 we pooled for each wavelength all data obtained and performed descriptive statistics (violin plots) on the displacement of the larva. In the new Figure 5—figure supplement 1 we provide the plots for all individual experiments. Both Figure 5 and Figure 5—figure supplement 1 provide evidence for a clear phototactic reaction.

Main changes in the text are in the subsection “Behavioral assays of *T. inopinata* larva”. Figure 5, Figure 5—figure supplement 1 and the respective figure legends are updated.

Figure 2: The evidence for xenopsins and cnidopsins forming a clade still finds low to moderate support only and might therefore represent an artifact. The possibility remains that Xenopsins exist in Lophotrochozoans only and this should be mentioned and visualized in the model.Even though many nodes in the xenopsin tree are well supported, it seems to suggest the existence of two paralogs in Lophotrochozoa that are distinctly present in molluscs, annelids and platyhelminths. This duplication appears to have occurred at the Lophotrochozoan root. If so, then the ciliary location would only be shown for one representative of one of the paralogs. This is important and should be mentioned.

To address the comments related to opsin evolution – the relation of xenopsins and cnidops and the diversification of xenopsins – we rerun the opsin tree with an improved taxon sampling by adding the recently published xenopsin sequences from the molluscs, *Sepia officinalis* and *Ambigolimax valentianus*, the bryozoan *Bugula neritina*, the flatworm *Maritigrella crozieri*, and the chaetognath *Pterosagitta draco*.

Figure 2 and all figure supplements of Figure 2 were updated and are based on the new inferred trees. Figure 2—figure supplements 5-8 are new and show trees of only xenopsins with different small outgroups. Main changes in the text are in the subsections “Behavioral assays of *T. inopinata* larva”, “Xenopsin is expressed in cilia of the eye photoreceptor cells in larval *T. inopinata*”, and “Evolution of bilaterian eye PRCs”.

In the new maximum-likelihood tree (IQ-TREE), for the clade composed of protostome xenopsins and cnidops, we find support values of ultrafast bootstrap and SH-like approximate likelihood ratio test ≥ 0.9 and approximate Bayes test ≥ 0.98. In a parallel Bayesian (Phylobayes) analysis this clade is supported with a posterior probability ≥ 0.95. We regard this support as being fairly strong and we show now in Figure 2 all four support values. High support values for the same clade were also reported by Rawlinson et al., 2019, and Ramirez et al., 2016. The latter regard cnidops even as being part of xenopsins. Nonetheless, we are interpreting this result carefully since the basal branching pattern of metazoan opsins differs considerably between recent studies on opsin phylogeny. Though opsin subgroups such as ciliary opsins, xenopsins, r-opsins, tetraopsins, etc. are confirmed by many studies, the interrelationships obviously depend very much on taxon sampling, choice of outgroup and tree inference algorithms used. Unfortunately, gene structure data cannot solve the question, whether cnidops and xenopsins are sister groups, since cnidops do not contain introns. Thus, we prefer to apply the term xenopsin only for protostome sequences as we did already in Vöcking et al., 2017, and as it was done by Rawlinson et al., 2019. Irrespective of these considerations, we interpret our data and published opsin phylogenies in favor for an emergence of xenopsins latest in the stem lineage of Bilateria and not lophotrochozoans. Otherwise we would expect a sister group relationship of xenopsins to another lophotrochozoan/protostome-specific opsin clade or a clade only composed of deuterostome opsins. This is not supported by our tree nor by our gene structure data and we are not aware that any publication provides respective evidence. In difference, all other protostome opsin sequences known form part of well-supported groups like r-opsin, tetraopsins, ciliary opsins and peropsins etc. In our view the simplest explanation of published and our gene tree and gene structure data is to assume a secondary loss of xenopsins in the lineage leading to deuterostomes.

We agree to the reviewers that the topology of the xenopsin subtree suggests an early divergence into two groups. This has been suggested also by Vöcking et al., 2017, and Rawlinson et al., 2019, though in both cases with moderate or low support. We did not follow this up in the first version of the manuscript since the support for the diversification of xenopsins into the two clades xenopsin A and xenopsin B was again not high. But we agree with the reviewer that this question is relevant for the manuscript. Accordingly, we did further analyses and discuss this aspect in more detail now. The revised analysis with the increased taxon-sampling did not change the picture. Support for the split is still lower as for instance for xenopsin as a whole or any other large opsin group. Thus, we run in addition trees of 1) xenopsins only (unrooted) and trees with 2) few cililary opsins, 3) few cnidops and 4) few ciliary opsins and cnidops as outgroup to investigate how robust the clades xenopsin A and xenopsin B against changes in the outgroup. The split between xenopsin A and xenopsin B is supported in the unrooted tree and in the tree rooted with cnidops, but xenopsin B has low support in the tree rooted with cnidops and ciliary opsins and is paraphyletic in the tree rooted only with ciliary opsins. We are discussing these findings in the revised version of the manuscript (subsection “Molecular phylogeny of animal xenopsins and c-opsins”) stating that our data suggest an early diversification of xenopsins is likely, but with moderate support only.

The xenopsin sequence of *M. fuliginosus* groups with xenopsin B representatives in all trees. We now mention this clearly in the manuscript (subsection “Molecular phylogeny of animal xenopsins and c-opsins”) and *M. fuliginosus* xenopsin may indeed be the first known xenopsin B of an annelid and the first xenopsin B, for which cellular expression data exist. In addition we compare sequence motifs which are important for G-protein coupling between *M. fuliginosus* xenopsin B and several flatworm xenopsin B, which may not be capable to induce G-protein signaling.

Since data on protein localization exist only from members of xenopsin A, we discuss considerations on the possible targeting of *M. fuliginosus* xenopsin more carefully now (subsection “Evolution of bilaterian eye PRCs”). Nevertheless, the closest related opsins of *M. fuliginosus* xenopsin for which data on subcellular localization exist, are still those, which enter cilia.

For Malacoceros xenopsin, the reasoning that xenopsin locates to cilia does not work out because the two rhabdomeric photoreceptors that strongly express it do not have cilia (only basal bodies). Where would the opsin go? There is no evidence for 'rudimentary cilia', they are simply absent (no acetylated tubulin staining). This should be stated as is.

Basal bodies close to the apical surface of PRCs are in the literature often interpreted as remnants of formerly well-developed cilia and the term rudimentary cilia is common in this context. We agree that the term is suggestive and we avoid it now. We describe the ultrastructure now more precisely (subsection “Xenopsin is coexpressed with r-opsin in cerebral eye PRCs in larval *M. fuliginosus*”, Figure 6—figure supplement 1) and we discuss this aspect now in more detail (subsection “Evolution of bilaterian eye PRCs”). We leave it open, where the xenopsin is going in the eye PRCs of *M. fuliginosus*. It may enter cilia in those cells, which have a prominent cilium and remain in the apical area of the plasma membrane in cells with reduced cilia or it may even enter microvilli. In the latter case xenopsin would be the first known opsin group, which is capable to enter both cilia and microvilli. However, all existing data on subcellular localization of xenopsin protein so far only show that xenopsin is able to enter cilia.

The figures, supplementary figures and figure legends seem to have been mixed in the new version as compared to the previous version. The numbers had disappeared. This confusing mess must be sorted out.

We are very sorry for the inconvenience and sorted the figures appropriately.